# The Enigmatic Role of TP53 in Germ Cell Tumours: Are We Missing Something?

**DOI:** 10.3390/ijms22137160

**Published:** 2021-07-02

**Authors:** Margaret Ottaviano, Emilio Francesco Giunta, Pasquale Rescigno, Ricardo Pereira Mestre, Laura Marandino, Marianna Tortora, Vittorio Riccio, Sara Parola, Milena Casula, Panagiotis Paliogiannis, Antonio Cossu, Ursula Maria Vogl, Davide Bosso, Mario Rosanova, Brunello Mazzola, Bruno Daniele, Giuseppe Palmieri, Giovannella Palmieri

**Affiliations:** 1Oncology Unit, Ospedale del Mare, 80147 Naples, Italy; davidebosso84@gmail.com (D.B.); rosanovamario@hotmail.com (M.R.); b.daniele@libero.it (B.D.); 2CRCTR Coordinating Rare Tumors Reference Center of Campania Region, 80131 Naples, Italy; marian.tortora@gmail.com (M.T.); giovpalm@unina.it (G.P.); 3IOSI (Oncology Institute of Southern Switzerland), Ente Ospedaliero Cantonale (EOC), 6500 Bellinzona, Switzerland; ricardo.pereiramestre@eoc.ch (R.P.M.); Laura.Marandino@eoc.ch (L.M.); Ursula.Vogl@eoc.ch (U.M.V.); 4Oncology Unit, Department of Precision Medicine, Università Degli Studi Della Campania Luigi Vanvitelli, 80131 Naples, Italy; emiliofrancescogiunta@gmail.com; 5Interdisciplinary Group for Translational Research and Clinical Trials, Urological Cancers (GIRT-Uro), Candiolo Cancer Institute, FPO-IRCCS, Candiolo, 10160 Turin, Italy; pasquale.rescigno@ircc.it; 6Department of Clinical Medicine and Surgery, Università degli studi di Napoli Federico II, 80131 Naples, Italy; vittorioriccio1990@gmail.com (V.R.); saraparola3@gmail.com (S.P.); 7Institute of Genetics and Biomedical Research (IRGB), National Research Council (CNR), 07100 Sassari, Italy; milena.casula@cnr.it (M.C.); gpalmieri@yahoo.com (G.P.); 8Departments of Biomedical Sciences and Medical, Surgical, Experimental Sciences, University of Sassari, 07100 Sassari, Italy; ppaliogiannis@uniss.it (P.P.); cossu@uniss.it (A.C.); 9Department of Urology, Ente Ospedaliero Cantonale (EOC), 6600 Locarno, Switzerland; Brunello.Mazzola@eoc.ch

**Keywords:** germ cell tumour, testicular cancer, TP53, thyroid cancer, Li-Fraumeni syndrome, somatic mutations, germline mutations, cisplatin resistance, cancer predisposing syndrome, bilateral testicular cancer

## Abstract

The cure rate of germ cell tumours (GCTs) has significantly increased from the late 1970s since the introduction of cisplatin-based therapy, which to date remains the milestone for GCTs treatment. The exquisite cisplatin sensitivity has been mainly explained by the over-expression in GCTs of wild-type TP53 protein and the lack of TP53 somatic mutations; however, several other mechanisms seem to be involved, many of which remain still elusive. The findings about the role of TP53 in platinum-sensitivity and resistance, as well as the reported evidence of second cancers (SCs) in GCT patients treated only with surgery, suggesting a spectrum of cancer predisposing syndromes, highlight the need for a deepened understanding of the role of TP53 in GCTs. In the following report we explore the complex role of TP53 in GCTs cisplatin-sensitivity and resistance mechanisms, passing through several recent genomic studies, as well as its role in GCT patients with SCs, going through our experience of Center of reference for both GCTs and cancer predisposing syndromes.

## 1. Introduction

Testicular germ cell tumours (TGCTs) are the most prevalent solid tumours in young men aged 15–35 years, with a rising incidence among Caucasians, although they represent overall 1% of all malignancies in men worldwide [1]. It has been estimated that 23,000 new cases of TGCT in Europe will be diagnosed each year by 2025, with an increase of 24% compared with 2005 [2]. The most reassuring data are that TGCT is one of the most curable solid cancers, with approximately 95% of men surviving at 5 years [3]. This percentage rises to 98–99% for patients diagnosed with clinical stage I (CSI) [4]. The cure rate has significantly increased from the late 1970s since the introduction of cisplatin-based therapy in the [5,6], which to date remains the milestone of germ cell tumours (GCTs) treatment. The exquisite cisplatin sensitivity has been for a long time mainly explained by the over-expression of wild-type TP53 protein and the lack of TP53 somatic mutations; however, several other mechanisms seem to be involved, many of which remain still elusive [7]. Nevertheless, approximately 10–15% of patients will develop a tumour relapse after initial treatment or a refractory disease [8] and non-seminomas, particularly those of extra-gonadal origin, are associated with a lower rate of platinum responses [9,10]. In recent years, these issues have boosted the question of platinum resistance in GCTs, which represents an ongoing unmet clinical need [11]. Another urgent issue, still not completely clarified and a direct consequence of cisplatin-related survival increase, is represented by the incidence of second malignancies in GCTs survivors [12]. As consequence, national prevention and long-term follow-up policies for survivors have been promptly promoted worldwide [13]. Indeed, several studies have reported a 1.7- to 3.5-fold increased risk for both haematological and solid second cancers (SSCs) in GCT survivors compared with age-matched general populations [14,15,16]. Historically, the risk of second cancers (SCs) has been associated with both radiotherapy (RT) and chemotherapy (CT), but not with surgery alone. Both cisplatin-based CT and therapeutic radiation for GCTs have clinically meaningful implications in the short as well as in the long term, with the occurrence and severity of these adverse events depending on the cumulative dose of CT [17]. Three recent reports have evaluated SC risk after cisplatin-based CT in GCT survivors, reporting a 40–80% excess risk [18,19,20]. Notably, the recent publication by Hellesnes et al. reported that SSC risk increased after surgery alone (standard incidence ratio SIR 1.28), with site-specific increased risks of thyroid cancer (SIR 4.95) and melanoma (SIR 1.94) [21], suggesting that a genetic susceptibility and/or that environmental elements during foetal life or early childhood might predispose for both GCT and other neoplasms [22,23,24]. However, the study by Hellesnes is lacking both germline and somatic mutations data, and, to date, GCT has not been associated with any defined cancer-predisposing syndrome, with only a few case reports focused on this issue being available. The findings about the role of TP53 in cisplatin-sensitivity and resistance, as well as the reported evidence of SSCs after only surgery in GCT patients, suggesting the spectrum of cancer-predisposing syndromes, highlight the need for a deepened understanding of the role of TP53 in GCT, with special attention to the “TP53-related inherited cancers”. In the following report we explore the complex role of TP53 in GCTs cisplatin-resistance mechanisms, going through several recent genomic studies, as well as the potential role of TP53 in determining SCs in GCT patients with SCs, as part of our experience as a Center of reference for both GCTs and cancer predisposing syndromes.

## 2. Platinum Resistance and Sensitivity in Germ Cell Tumours

### 2.1. General Overview

Several extensive genomic studies have tried to identify pre-target, on-target, and post-target factors that could explain the two sides of the same coin: the exquisite sensitivity of GCTs to platinum-based CT and, at the same time, drug resistance and consequent poor survival when these factors are lacking [25,26]. TGCTs are characterized by significant chromosomic defects, such as aneuploidies. Indeed, as already acknowledged, the gain of chromosome arm 12p, along with high rates of copy number changes [27], are considered the most common chromosomic alterations of GCTs. Somatic mutations, instead, have very low frequency, with the most common activating mutations from diagnosis being in the following 3 genes: KIT (18%), KRAS (14%), and NRAS (4%), and thus reported as driver truncal events [28,29].

Nucleotide excision repair (NER) is the main DNA repair system involved in platinum-induced adducts resolution and comprises approximately 30 proteins involved in DNA damage recognition, incision/excision, gap filling, and ligation [30]. Consequently, disruptions in the NER system cause a lack of DNA repair, adding to damage accumulation, which finally increases apoptosis and could explain responsiveness to cisplatin in TGCTs [30,31,32]. Moreover, single-strand breaks can result in double-strand breaks that are fixed, under normal conditions, by the homologous recombination (HR) system, of which BRCA2 and PARP are essential components. Defects in the HR system can also explain the sensitivity to cisplatin [33]. Although somatic mutations in BRCA1/2 are not common in TGCT, the mutational signature associated with HR defects and high frequency of inactivation by methylation of BRCA1 and RAD51C have been reported in these tumours [28,34,35]. More controversial is the role of the tumour mutational burden (TMB). In a recent analysis considering whole exome sequencing (WES) from 290 TGCTs, the mean nonsynonymous TMB was 0.33 Mb (range 0–9.4 Mb) [34], considerably lower than that of other adult solid tumours. In the multifactorial analysis, adjusting for purity, TMB was significantly higher in platinum-resistant than in sensitive TGCTs, with a mean increase of ~0.35 nonsynonymous mutations per megabase [34].

Furthermore, several studies have explored the association between epigenetic factors, such as DNA methylation and platinum resistance. Hypermethylation is able to inhibit transcription of CpG-rich promoter regions of tumour suppressor genes, thus leading to gene silencing [36]. Compared with other solid tumours, TGCTs are characterized by a global CpG hypomethylation [37,38]. Shen et al. showed that seminoma, embryonal carcinoma, yolk sac tumour, and teratoma histologic subtypes harboured distinct DNA methylation profiles [28]. Consistent with previous reports, seminomas, which are generally considered more platinum sensitive, are severely hypomethylated, while teratomas, yolk sac tumours, and choriocarcinomas have the highest level of DNA methylation [28,39,40,41,42].

### 2.2. The Role of TP53 and Its Pathway

Mutations/losses of the tumour-suppressor TP53 and amplification/gains of its regulator, MDM2, as well as WNT/CTNNB1 pathway aberrations, which are involved in developmental processes and stemness, have been deeply investigated in platinum-resistant and metastatic disease. As already acknowledged, one of the hallmarks of GCTs is the presence of wild-type (WT) *tp*53 in most patients, although mutations in *tp*53 have been described in a few cases in the past [43] and more recently in a small subset (~7–15%) of cisplatin-resistant or relapsed GCT patients [34,44]. Activity of TP53 is regulated by the E3 ubiquitin ligase MDM2, which inhibits transcription of target genes by TP53 within direct binding to the transactivation domain of TP53 and through targeting TP53 for proteasome degradation by ubiquitination [45,46]. In physiological conditions, MDM2, through an auto-regulatory negative feedback, keeps TP53 levels low. Genomic alterations affecting both TP53 and MDM2 have been described in resistant GCTs [47]. A large study assessed 180 GCTs using whole exome sequencing, identifying MDM2/TP53 alterations exclusively in cisplatin-resistant tumours [44]. Notably, mutually exclusive TP53/MDM2 alterations were significantly more frequent among patients with unfavourable clinical characteristics, according to the IGCCCG poor-risk group [48] and having a mediastinal non-seminoma primary tumour site. The majority of MDM2 amplifications were observed in post-treatment samples, suggesting that tumour cells are selected during treatment [44]. Since *tp*53 mutations are rare, MDM2 amplification is a possible selection mechanism to prevent cell cycle arrest and DNA repair during the progression of disease. This hypothesis is reinforced by functional studies within cisplatin-sensitive and resistant testis cancer cell lines, indicating that the interaction between TP53 and MDM2 needed higher doses of cisplatin to be disrupted in resistant ones [49]. As previously reported, although most GCTs express WT TP53, posttranscriptional modifications are able to suppress the pro-apoptotic activity of TP53. For example, lysine methylation at the carboxyl terminus of TP53 represses its transcriptional activity, as demonstrated by reduced expression of PUMA and p21 in TC cells [50]. Furthermore, the expression of miR-372 and miR-373 was reported to block TP53 signalling, and their elevated expression levels have been detected in cisplatin-resistant GCT cell lines [51,52,53]. Another posttranslational modification is represented by deacetylation of TP53 by SIRT1, which may potentially repress TP53 activity in TGCT, suggesting that SIRT1 acts as an oncogene. However, the exact role of SIRT1 impacting TP53 activity in the context of TGCT has not be yet completely clarified since it has been reported as upregulated in various malignancies and acting as a tumour suppressor in other reports [54]. Moreover, the extrinsic apoptosis pathway via the interaction between FAS and FAS ligand is also activated by cisplatin, which increases expression of the FAS death receptor, a transcriptional target of TP53 [49,55]. Other studies presented the use of cisplatin results in an increased expression of pro-apoptotic proteins PUMA and NOXA, which are involved in the intrinsic apoptosis pathway [56,57,58]. Additionally, high protein levels of the pluripotency factor OCT4, typical for the embryonal carcinoma histotype, are associated with a high expression of NOXA, while OCT4 knockdown resulted in NOXA decrease [58]. Notably, high NOXA expression has been linked to TGCT, with good prognosis and embryonal carcinoma histology [59]. Crosstalk between the extrinsic and intrinsic apoptosis pathways, where FAS receptor activation results in caspase-8-mediated cleavage of BID, may further reinforce the apoptotic response. In a comprehensive genomic study, Taylor-Weiner et al. showed that primary TGCTs are uniformly wild type for TP53 and, by functional measurement of apoptotic signalling (BH3 profiling), they have demonstrated high mitochondrial priming that facilitates chemotherapy-induced apoptosis [60]. (Figure 1).

## 3. Germ Cell Tumours and Second Cancers

### 3.1. General Overview

An increased risk of second malignancies in cancer survivors has been associated with previous exposure to CT and/or RT [61]. In GCT survivors, an increased risk of various second tumours, comprising haematological, gastro-intestinal (GI), and urologic malignancies is already well established [62,63,64,65,66,67,68]. A significant 3-fold risk of leukaemia, after a median latency of 5 years, was reported in an international population-based study of 18,576 TGCTs who underwent pelvic RT [69], and both cisplatin and etoposide have been associated with elevated risks of secondary leukaemia in survivors [69,70]. Alkylating agents, such as cisplatin, have been associated with leukaemia diagnosis 5 to 8 years after CT, usually preceded by myelodysplastic syndromes (MDS) with a karyotype involving long-arm deletions or monosomy of chromosomes 5 and 7 [71]. Topoisomerase II inhibitors, such as etoposide, might be responsible for leukaemia after a median onset of 2 to 3 years, usually without MDS and often characterized by balanced translocations involving the MLL (11q23), RARA (17q21), and RUNX1 (21q22) loci [71]. A significant dose–response relationship between cumulative cisplatin dose and leukaemia risk, after adjusting for RT dose, was registered in a series of 18,000 TGCT survivors, after a median follow-up of 10.2 years, where cisplatin dose of 650 mg was associated with a significant 3.2-fold risk of leukaemia, with higher doses (1000 mg) associated with significant 6-fold risks [69]. The 5-year cumulative incidence of leukaemia after cumulative etoposide doses of 2000 and >2000 mg/m2 was 0.5% and 2.0%, respectively. A Danish nationwide cohort study of 5,190 TGCT described a significant 6.3-fold risk of myeloid leukaemia after combined chemotherapy with cisplatin, etoposide, and bleomycin (BEP scheme) (*p* < 0.001) compared with age-matched population-based controls (*n* = 56 myeloid leukaemia for the entire cohort) [63]. Regarding the risk of SSCs and RT, this was significantly higher in anatomical sites within infra-diaphragmatic RT fields (relative risk (RR), 2.7; *n* = 5212) compared with unexposed sites (RR, 1.6), persisting elevated for more than 35 years, as reported in an international population-based analysis of 40,576 10-year TGCT survivors, [72]. A significant 5.9-fold risk of stomach cancer, with >20-fold risk if gastric radiation doses of >50 Gy (8 cases vs. 6 controls) vs. 10 Gy (15 cases vs. 49 controls) was reported in an analytic study of 5-year TGCT survivors [73]. Another multicentre study reported that the hazard ratio (HR) of an infra-diaphragmatic SSC increased by 8% per Gy of radiation dose delivered (*p* < 0.001) compared with patients who received no para-aortic RT [20]. However, it must be remarked that most of these series explored the prevalence of SCs in GCT survivors before the adoption of cisplatin-based CT [62,72,74,75]. More recently, three studies reported significant associations of SSCs risk in TGCT survivors related to cisplatin-based CT [18,19,20]. The first study investigated the risk of SSC in 12,000 USA TGCT patients treated with cisplatin-based CT over a 30-year period, reporting an overall 1.4-fold significant risk of SSC (*n* = 5111) compared with surgery-only patients. Significant risks of kidney (SIR 3.4; *n* = 513), thyroid (SIR, 4.4; *n* = 511), and soft tissue cancers (SIR, 7.5; *n* = 510) were registered [18]. The Danish population-based study of 5190 TGCT survivors, after a median follow-up of 14.4 years, reported a SC risk after BEP of 1.7-fold higher (95% CI 1.4–2.0) compared with surveillance, with a 20-year cumulative SC incidence of 7.6% after BEP [19]. Analogously, the Dutch hospital-based study, after a median follow-up of 14.1 years, registered a significant 2.4-fold risk of SC in BEP-treated TGCTs (*n* = 5151) compared with those not treated with CT. The authors highlighted a significant linear dose-dependent increase in the risk of GI malignancies by 53% with each additional 100 mg/m2 of platinum-containing CT (*p* ≤ 0.001), thus supporting evidence of a potential dose-dependent relationship between platinum-based CT and SCs. The risk of GI SCs was increased by 3.6-fold (*n* = 521) and 5.0-fold (*n* = 516) after 400 to 499 and >500 mg/m2 of platinum-containing chemotherapy, respectively [20]. Different from the above-mentioned studies, an analysis of data on Norwegian men diagnosed with TGCT between 1980 and 2009 showed, instead, an increased overall risk for SCs among survivors, despite treatment [21]. Median observation time for the total cohort was 16.6 years, and 37% of survivors had an observation time >20 years. Risk was elevated in particular beyond 10 years of follow-up after cisplatin-based CT or RT. Despite reduced treatment intensity, two or more cycles of cisplatin-based CT were associated with continuing increased SC risk. Overall, 572 TGCT survivors (10.2%) developed 651 SCs, with prostate, lung, bladder, melanoma, and colon cancer being the most common malignancies with a 58% overall excess risk of developing non-germ cell SC (SIR 1.58, 95% CI 1.45–1.71) compared with the general population. All treatment groups had significantly increased risks, ranging from 28% excess risk after surgery only to 2-fold increased risk after cisplatin plus RT. Intriguingly, after surgery only, there were increased risks for melanoma (SIR 1.94, 95% CI 1.10–3.42) and thyroid cancer (SIR 4.95, 95% CI 1.86–13.18). Cisplatin was associated with a significant 1.9- to 3.7-fold increased risk of cancers of the small intestine, lung, melanoma, kidney, and bladder, in line with previous reports. After RT, the risks were 1.5–4.4 times significantly increased for cancers of the stomach, small intestine, liver and bile ducts, pancreas, lung, kidney, and bladder. The association of both treatments increased the risks for cancers of the stomach, small intestine, pancreas, soft tissue, thyroid, lymphoma, and leukaemia [21].

### 3.2. TGCT, Thyroid Cancers, and TP53: Experience from a Reference Center

Since the elevated risk of thyroid cancer (TC), reported for the surgery only group in the study by Hellesnes et al. [21], is a novel finding that needs to be further investigated, we carried on a database search of TGCT patients, referred to the Rare Tumors Reference Center of Campania Region between 2000 and 2019, who received a TC diagnosis. Finally, we identified two patients, who received a diagnosis of TGCT and metachronous TC (Figure 2 and Figure 3). Both patients had no family cancer history that was suspicious for hereditary cancer predisposing syndrome or professional exposure. The first patient had a history of metachronous bilateral stage I seminoma, managed with surveillance (surgery only) and a subsequent diagnosis of papillary thyroid cancer (PTC), 6 years from the first TGCT diagnosis and 1 year after the second one (Figure 2). The second patient had a history of left cryptorchidism and a diagnosis of stage I right-sided seminoma treated with radiotherapy in an adjuvant setting. He received a diagnosis of follicular thyroid cancer (FTC) 5 years after the end of treatment (Figure 3). Both patients are still alive and free of disease relapse (clinical details are reported in Figure 2 and Figure 3). Biopsy tissue samples of TGCT, TC, and, for the first case, also tissue sample from neck lymph node, were collected and analysed for somatic mutation analysis (Methods section is provided in the Appendix A). At the time of this analysis, both patients, in absence of a suspicious cancer family history, refused germ-line testing.

We identified several genetic variants in both types of tumours, some of them neutral or of uncertain significance, as well as three pathogenic mutations confirmed by pyrosequencing in the BRAF, KRAS, and TP53 genes. Figure 2 and Figure 3 summarize the genetic alterations found in our cases. The BRAF V600E mutation was detected in the thyroid tumour of patient 1, while the KRAS G12S in the testicular seminoma of patient 2. The most common neutral/uncertain mutations detected involved the TP53 and PIC3CA genes in both thyroid and testicular cancers. The c.216delC pathogenic mutation, located in coding exon 3 of the TP53 gene, results from a deletion of one nucleotide at nucleotide position 216, causing a translational frameshift with a predicted alternate stop codon (p.V73Wfs*50). This alteration is expected to result in loss of function by premature protein truncation or nonsense-mediated mRNA decay. As such, this alteration is interpreted as a disease-causing mutation and has been found in thyroid, node, and left testicular cancers of patient 1 and in TC of patient 2.

The association of TC and TGCT has been previously reported in both epidemiological studies and case reports [18,21,76]. TC accounts for approximately 2% of all cancers diagnosed worldwide and is the most common malignant endocrine neoplasm [77,78]. Several studies evidenced a continuously increasing incidence of TC in several geographic areas in the last decades [79,80]. TC types are classified according to their histological features. The differentiated TC (DTC), deriving from follicular cells and sub-classified as papillary thyroid carcinoma (PTC) and follicular thyroid carcinoma (FTC), is the most common histotype, representing approximately 90% of all TCs [81]. Anaplastic thyroid carcinomas are rarer but more aggressive neoplasms [81]. Hereditary thyroid neoplasms arising from calcitonin-producing C cells are known as familial medullary TC and include well-documented syndromes such as multiple endocrine neoplasia IIA or IIB and pure familial medullary thyroid carcinoma syndrome. The most recognized common risk factor of TC is exposure to radiation, either from environmental, medical, or other sources. However, about 5–15% of DTC cases are thought to be of familial origin [82]. The clinical features of familiar DTC are controversial due to the high genetic heterogeneity and are sub-classified according to clinicopathological correlations into two groups. Among the first group, included syndromes are characterised by a predominance of non-thyroidal tumours, including familial adenomatous polyposis, Cowden syndrome, Werner syndrome, Carney complex, and Pendred syndrome. The second group comprises a spectrum of familial syndromes characterised by a predominance of non-medullary TC, such as pure familial papillary thyroid carcinoma with or without oxyphilia, familial papillary thyroid carcinoma with papillary renal cell carcinoma, and familial papillary carcinoma with multinodular goitre. Most familial TCs have been described as being more aggressive than sporadic ones, with a predisposition for lymph node metastasis, extrathyroidal invasion, and a younger age of onset [83,84]. More controversial and still under investigation is the role of TC as potential component of the Li-Fraumeni syndrome (LFS). In a review of a French cohort including 415 LFS carriers with classic DNA-binding domain mutations in the *tp53* gene, TC was present in only 0.9% of the patients [85]. However, a retrospective study including 101 Brazilian p.R337H TP53 mutation-carrying family members with cancer, reported 11 cases of TC (10.9%), highlighting that, although TC is an uncommon manifestation of LFS in general, this malignancy appears to be a component of the spectrum of tumours in carriers with the TP53 p.R337H founder mutation [86]. In this regard, it is important to emphasise that LFS represents a paradigm in genetic predisposition to cancer, considering the pivotal role of TP53 in the response to DNA and that its germline mutations are the fundamental to diagnose LFS, which has considerably evolved since its original definition in 1969 by Frederick Li and Joseph Fraumeni [85,87,88,89,90,91,92,93]. Indeed, a notable feature of the LFS is the heterogeneity of its clinical presentation in terms of tumour type and age of tumour onset. Despite germline TP53 mutations being identified in familial aggregations of childhood and adult tumours, they can also be detected in patients and families who have developed only adult cancers and in patients without a familial history of cancer [85,94,95]. To date, according to our knowledge, no association of both TGCT and TC has been reported among LFS carriers. Moreover, in the survivorship reports this association is mainly attributed to previous delivered treatments. In the report by Travis et al., 30 of 40,576 long-term survivor TGCT patients had TCs [72], though the observed expected ratio of TCs in this study was not higher than those observed in other common malignancies. Nevertheless, the study demonstrated a relative risk of 3.1 (95% CI 1.2–6.7) for TC development after radiotherapy treatment alone, a result higher than those reported in older articles [96]. This is surprising considering that newer radiation fields expanded less than older treatments. In the study by Fung et al., the patients were treated initially with either CT (*n* = 6,013) or surgery (*n* = 6,678) but not RT [18]. The authors reported again significantly increased risks for kidney, thyroid (4-fold increase), and soft tissue tumours, highlighting the increased risk of SSCs among patients with testicular non-seminoma treated with cisplatin-based CT. Veiga et al. performed an epidemiological study focusing on TC in patients treated with CT and/or RT for childhood cancers [97]. The authors showed that 30 years after the first childhood cancer treatment, the cumulative incidence of TC was 1.3% (95% CI 1.0–1.6) for females and 0.6% (95% CI 0.4–0.8) for males. Furthermore, among patients who received thyroid radiation doses ≤20 Gy, treatment with alkylating agents was associated with a significant 2.4-fold increased risk of TC (95% CI 1.3–4.5), while no CT-related risk was evident for thyroid radiation doses >20 Gy. Globally, these studies suggest that both RT and CT, alone or in combination, may promote TC. Spiliopoulou et al. have described three detailed cases of metachronous TC in patients with a previous diagnosis of TGCT [76]. As in our series, all the patients were young at the time of their diagnosis (range 27–42 years); they had PTC and non-seminoma in all the cases, while in our series one FTC was found, and seminoma was diagnosed in both testis of patient 1 and also in patient 2. Furthermore, in the Spiliopoulou case series, after orchiectomy all patients received BEP, albeit at different cumulative doses. The diagnosis of TC was made at different time intervals: in one case within 5 months of completion of CT, whereas in the remaining two cases after approximately 12 and 5 years. The authors wondered if there was potential involvement of inherited syndromes, which can cause both thyroid and testicular tumours, such as the Carney Complex and the Cowden syndrome, but no evidence of their involvement in the published cases exists. Overall, it is currently difficult to establish whether the association between TC and previous TGCT is real or if it depends on the constantly increasing incidence of TC in combination with modern improvements in the early detection of thyroid neoplasms, especially in a subgroup of patients undergoing oncological follow-up. In addition, the long survival reached in GCT patients allows the arousal of further tumours, especially those with increasing incidence in the general population. Some authors hypothesized that CT or RT delivered for the treatment of GCTs might cause tumours of the thyroid gland, but this hypothesis does not contemplate the increased risk of TC also in TGCT treated with surgery only, as reported in the series by Hellesnes and as shown in our patient 1. This arouses the suspicion of specific genetic alterations that may underlay the pathogenesis of both these tumours. Indeed, despite our patients’ refusal of a germline test in the absence of a cancer family history, the finding of p.V73Wfs*50 TP53 somatic mutation in thyroid, node, and left testicular cancers of patient 1 and in TC of patient 2 is worthy of consideration. First of all, we identified a TP53 somatic mutation in specimens of both seminoma testis in patient 1, with a final diagnosis of stage I TGCT treated with surgery alone followed by surveillance, that to date, after 10 years of follow-up, has not yet relapsed. This aberration seems to be different from several previous reports of somatic TP53 mutations described in a small subset (~7–15%) of cisplatin-resistant or relapsed GCT and more frequently among patients with IGCCCG poor-risk group and having a mediastinal non-seminoma primary tumour site [34,44]. Secondly, the same p.V73Wfs*50 TP53 mutation has already been described in one study as germline mutation of a 15-year-old LFS carrier with osteosarcoma whose mother had breast cancer at age 25 [98]. In the study by Toguchida et al., they examined the possibility that some patients with sarcomas and no family history of cancer might be carriers of new TP53 germline mutations. Moreover, the authors wondered about the possibility that some patients with sarcoma whose personal or family tumour history suggested a predisposition to cancer, even in the absence of the criteria for the LFS, carried germline mutations of the TP53 gene. Finally, they concluded that new identified TP53 germline mutations, such as p.V73Wfs*50, are rare among patients with "sporadic" sarcoma but that they may be common in patients with sarcoma whose background includes either multiple primary cancers or a family history of cancer, thus identifying a group of patients with cancer who carry germline mutations of the TP53 gene more diverse than that suggested by the clinical definition of the LFS [98]. Despite the previous delivered treatments and the family history of cancers, also in the absence of a well-known suspicious familiar cancer predisposing syndrome, GCTs survivor patients with a diagnosis of SCs during their life should be promptly referred to genetic counselling.

## 4. Conclusions

Somatic mutations of TP53 are very rare in TGCTs, different from other malignancies, and when present they may notably impact prognosis and chemotherapy sensitivity. The potential role of TP53 in the diagnosis of SCs in TGCT survivorship needs to be further and more deeply investigated in the light of our preliminary findings. The identification of TP53 germ-line mutation carriers could be of substantial clinical importance, raising serious questions about appropriate methods of cancer surveillance and counselling for these persons.

Oncologists who follow up patients with a previous diagnosis of TGCT should always keep in mind the potential risk for a second primary TC, independent of the delivered treatments, since SCs may arise also in patients who were treated only with surgery.

## Figures and Tables

**Figure 1 ijms-22-07160-f001:**
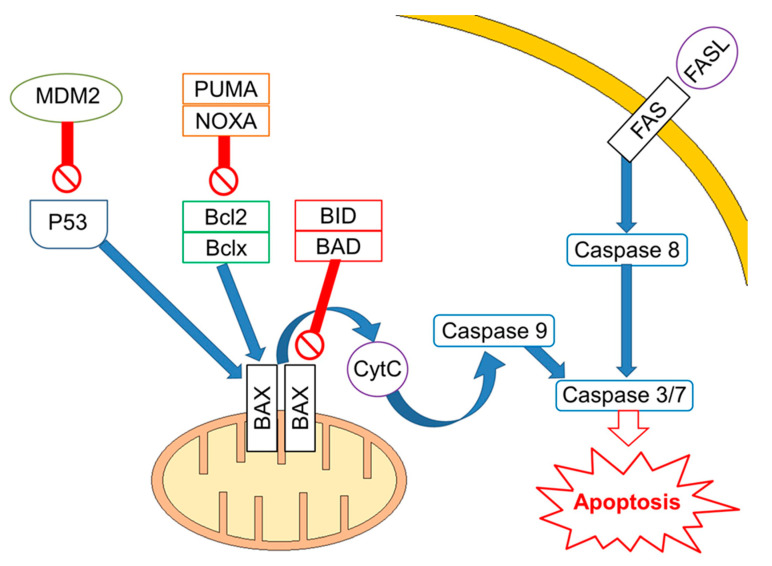
Apoptotic signalling in cisplatin-sensitive Germ Cell Tumours. Apoptosis is regulated at several levels with a key role for TP53, which transcriptionally regulates proteins involved in both the extrinsic and intrinsic apoptosis pathway, including FAS death receptor, BAX, PUMA, and NOXA. Activated TP53 enhances expression of the FAS death receptor. Binding of FAS ligand to FAS death receptor leads to cleavage of procaspase-8 and subsequently activates a caspase cascade leading to apoptosis. The intrinsic apoptosis pathway is regulated by pro- and anti-apoptotic proteins. Pro-apoptotic proteins directly facilitate BAX and BAK oligomerization in the mitochondrial outer membrane, or they indirectly inhibit anti-apoptotic proteins. Crosstalk between the extrinsic and intrinsic apoptosis pathways exists in form of caspase-8-mediated cleavage of BID.

**Figure 2 ijms-22-07160-f002:**
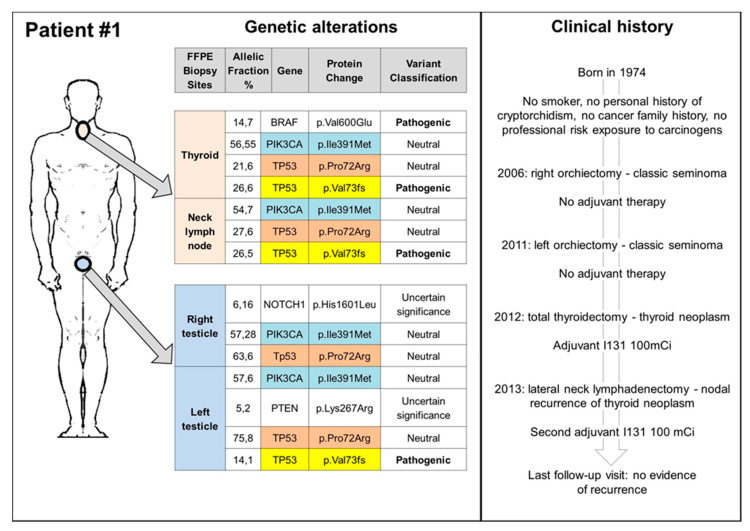
Clinical history and somatic genetic alterations of patient 1. FFPE: formalin-fixed paraffin-embedded.

**Figure 3 ijms-22-07160-f003:**
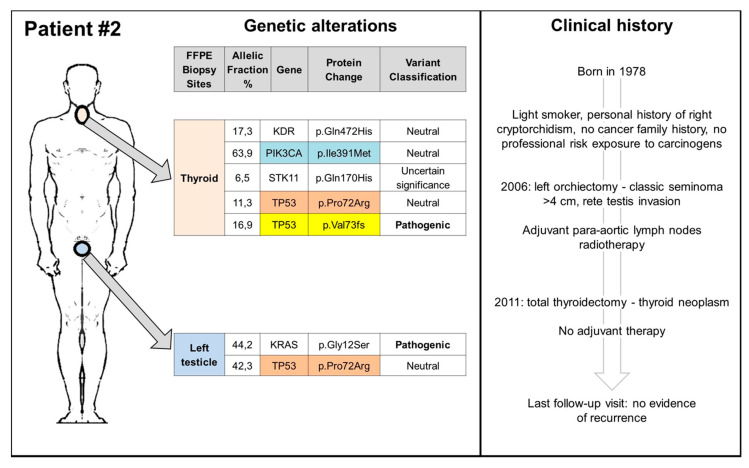
Clinical history and somatic genetic alterations of patient 2. FFPE: Formalin-Fixed Paraffin-Embedded.

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
