# Peer review of "The Enigmatic Role of TP53 in Germ Cell Tumours: Are We Missing Something?"

_ijms, 2021, doi:10.3390/ijms22137160_

Round 1

Reviewer 1 Report

In this review, the authors would like to discuss recent evidence about the role of TP53 in GCTs cisplatin-sensitivity and resistance mechanisms, considering recent genomic studies and their results obtained from center of reference for both GCTs and cancer predisposing syndromes.

The topic is interesting and relevant in the fields.

The oncogenic impact of p53 protein and the chemoresistance in germ cell tumors makes this review current and an important source of bibliographic information. It is well written and well structured in the rationale.

Paragraph 3.2, which describes the authors' experimental activity of the gruoup, well centers in the manuscript and is carried out good enough and informative from a methodological point of view. Authors should review the style of English to make reading smoother.

Overall, the manuscript is well centered and suitable for the publication.

Author Response

  • We greatly appreciate the First Reviewer's valuable comments and we thank the Reviewer for having grasped the main aim of our manuscript. Accordingly to the received suggestions, we edited the English style for making the reading smoother. Please, see the tracked changes version of the resubmitted manuscript.  

Reviewer 2 Report

The manuscript "The enigmatic role of TP53 in germ cell tumors: Are we missing something?" analyzes the outcomes of the current genomic and clinical research that explore the role of TP53 in cisplatin sensitivity and resistance mechanisms in germ cell tumors (GCT), and in emerging of second tumors in GCT patients treated with radiation and chemotherapy and surgery. The review provides critical and comprehensive analysis of possible involvement of the TP53 mutations and expression in initiation and progression of the GCT and second tumors, as well as their cisplatin-sensitivity and resistance and correlation with some cancer predisposing syndromes. The manuscript also presents an analysis of the TP53 mutations and their clinicopathological data for two cases in GCT patients with secondary thyroid cancer as example of new disease-causing mutation.

This article may be interesting to a wide circle of researchers and physicians. The information provided in this manuscript may be useful for further research. The manuscript gives an overview of the latest findings that well organized and comprehensively described. The references were used properly.

I have only minor comment to be addressed.

I believe that the following article relates to the review topic and should also be cited in the presented manuscript.

J.T. Lafin, A. Bagrodia, S. Wolduand, J. F. Amatrudaю New insights into germ cell tumor genomics. Andrology, 2019, 7, 507–515

Author Response

  • We would like to express our great appreciation to the second reviewer for her/him careful and precious observations on our manuscript. We have now revised our manuscript to address and accommodate the valuable reviewer’s suggestion. The manuscript’s bibliography has been updated including the suggested article. Please, see the tracked changes version of the resubmitted manuscript.